**www.cambridge.org/ext**

## Research Article

paleontology; community ecology; community assembly; mammals; climate change

**Corresponding author:**
Misha A.J.B. Whittingham;
Email: mishawhittingham@cmail.carleton.ca

# Functional stasis and changing habitat preferences among mammalian communities from the PETM of the Bighorn Basin, Wyoming

Misha A.J.B. Whittingham[1] ⬤, Vera A. Korasidis[2,3] and Danielle Fraser[1,3,4,5]

[1]Department of Earth Sciences, Carleton University, Ottawa, ON, Canada; [2]School of Geography, Earth and Atmospheric Sciences, University of Melbourne, Parkville, VIC, AUS; [3]Department of Paleobiology, National Museum of Natural History, Smithsonian Institution, Washington, DC, USA; [4]Paleobiology, Canadian Museum of Nature, Ottawa, ON, Canada and [5]Biology, Carleton University, Ottawa, ON, Canada

## Abstract

The transition between the Paleocene and Eocene epochs (ca. 56 Ma) was marked by a period of rapid global warming of 5 °C to 8 °C following a carbon isotope excursion (CIE) lasting 200 ky or less referred to as the Paleocene-Eocene Thermal Maximum (PETM). The PETM precipitated a significant shift in the composition of North American floral communities and major mammalian turnover. We explored the ecological impacts of this phenomenon by analyzing 173 mammal species from the Bighorn Basin, Wyoming, USA, including their associated body alongside a database of 30 palynofloral localities as proxies for habitat. For each time bin, we calculated mean and median differences in body mass and habitat preference between significantly aggregated and segregated mammal species. Aggregated species showed significant similarity in habitat preference only prior to the PETM, after which habitat preference ceased to be a significant factor in community assembly. Our measures of differences in body mass space provide no evidence of a significant impact of competitive interactions on community assembly across the PETM, aligning with previous work. Our results indicate the persistence of a stable mammalian functional community structure despite taxonomic turnover, climate change and broadening habitat preferences.

## Impact Statement

Here, we combine multiple measurements of niche occupation based on two different definitions of the niche concept to examine how terrestrial mammalian communities responded to climate change and range shifts in the past. We provide a framework for analyses of community paleoecology, which incorporates both environmental (i.e., habitat) and morphological estimates of niche occupation at a community scale. As a case study for this methodology, we examined mammalian communities from the Paleocene-Eocene Thermal Maximum. Our results show that, on intermediate timescales, the variation in functional traits exhibited between species in the same communities can be conserved, possibly enabled by increasing primary productivity. In the aftermath of climate change and introduction of immigrant fauna, habitat preferences broadened within communities and narrowed between communities. The results are communities dominated by immigrant taxa where habitat preference is roughly homogenous between segregated species, and no longer a factor in determining community assembly. Our findings echo observations of other modern and fossil range shift events, and, combined with our expanded methodology, may aid in predicting and understanding how mammalian communities respond to climatological perturbation in the absence of humans.

## Introduction

Anthropogenic climate change and species transplantation have driven significant changes in, and homogenization of, species composition (Vitousek et al., 1996; Vitousek et al., 1997; McKinney and Lockwood, 1999; Parmesan and Yohe, 2003; Olden et al., 2004; Parmesan, 2006; Chen et al., 2011; Fraser et al., 2022) as well as losses of functional diversity (Naeem et al., 2012; Mouillot et al., 2013; Matuoka et al., 2020; Li et al., 2022) among modern and fossil terrestrial communities. Human activities have also driven significant changes in mammal community structure in the form of changes in species associations (Lyons et al., 2016; Tóth et al., 2019; Pineda-Munoz et al., 2020). Given that neontological research is hindered by the typically short time frames available for studies of community response to anthropogenic disturbance, the fossil record allows us to readily examine the long-term ecological and evolutionary impacts of similar perturbations in the past (Dietl et al., 2015; Barnosky et al., 2017).

Understanding how communities respond to changes in climate, environment, community composition and species associations in the past can provide data on how modern communities may respond to the effects of ongoing anthropogenic change. In this context, one interval of highly comparable, though less rapid, climate change, is the Paleocene-Eocene Thermal Maximum (PETM) (Kennett and Stott, 1991; Koch et al., 1992; Zachos et al., 2005; Smith et al., 2009; Gingerich, 2019).

The PETM was a period of major climate change characterized by a large (5 °C–8 °C), temporary increase in global temperatures that resulted from the release of thousands of Pg of isotopically light carbon in a period of <10 ky, commonly known as the onset of the Carbon Isotope Excursion (CIE) (Kennett and Stott, 1991; Dickens et al., 1995). The main body of the CIE has been estimated to have a duration of ~115 ka, while the recovery from the excursion has been estimated to have an excursion of another ~42 ka (Aziz et al., 2008; McInerny and Wing, 2011). In the terrestrial realm, the PETM is particularly well represented in the Bighorn Basin of Northern Wyoming, which preserves both faunas, including mammals, and floras (Koch et al., 2003; Wing et al. 2005; Abels et al., 2016). Floral records from the Bighorn Basin show that the changing climate had profound impacts on plant communities; closed-canopy wetland floras were replaced by shrubs and trees characteristic of drier, more open environments (Wing et al., 2003; Wing and Currano, 2013; Korasidis et al., 2022a, 2022b). The mammal communities of the Bighorn Basin subsequently experienced turnover, coinciding with the onset of the PETM (Gingerich, 2003). Faunal turnover during the PETM has generally been interpreted as a mass immigration event, resulting in the local disappearance of plesiadapids and the arrival of Eurasian mammals, including primates, perissodactyls and artiodactyls (Bowen et al., 2002; Gingerich, 2006). The immigration of Eurasian mammals into North America and northward range shifts were accompanied by a sharp rise in species richness and evenness in the post-PETM communities (Clyde and Gingerich, 1998; Bowen et al., 2002; Burger, 2012). Rapid climate change and immigration during the PETM are also associated with transient dwarfing among several mammal genera (Gingerich, 2003; Burger, 2012; Secord et al., 2012) but minimal changes in community structure (i.e., phylogenetic community structure, body mass dispersion and beta diversity; Fraser and Lyons, 2020). Though the picture of ecological change during the PETM is becoming clearer, changes in mammal species associations and their comparative occupation of niche space have been little explored. We therefore ask: in terms of both ecological roles and environmental (i.e., habitat) preferences, are there observable changes in the niche occupation among co-occurring mammals across the PETM?

The basic conceptual models of community assembly and classical niche theory predict that community assembly mechanisms are reflected in the packing of coexisting species in niche space (MacArthur and Levins, 1967; MacArthur and Wilson, 1967; Hubbell, 2001). By quantifying changes in species associations and the ways those species divide niche space, we can better understand the drivers of ecological change. Traits (e.g., body mass, locomotor strategy and diet) are most often used as proxies for the functional role of a species (Bambach et al., 2007; Chen et al., 2019; Fraser and Lyons, 2020; Pineda-Munoz et al., 2020). Sharing and partitioning of niche space can be quantified using single functional trait axes (e.g., body mass; Pineda-Munoz et al. [2016]) or by constructing ecomorphospaces incorporating multiple functional trait axes (e.g., Bambach et al., 2007; Chen et al., 2019), both of which constitute estimates of functional richness (i.e., the total amount of occupied

niche space) following the definition by Legras et al. (2018) (e.g., Mason et al., 2005; Villéger et al., 2008). Similarly, calculations of total morphological disparity can be used as metrics for niche overlap (e.g., Bapst et al., 2012; Whittingham et al., 2020). Changes in the degree of overlap in trait space, whether univariate or multivariate, among coexisting species are then interpreted as changes in niche similarity, reflecting underlying assembly processes such as limiting similarity (MacArthur and Levins, 1967), competitive exclusion (Hardin, 1960) and environmental or habitat filtering (Soininen et al., 2007a, 2007b; Soininen, 2010). Given the abiotic and biotic changes that typified the PETM, we expect considerable changes in the ways coexisting species divided available niche space.

Herein, we analyze changes in species associations using the methods of Ulrich (2008), which involves the calculation of co-occurrence metrics (i.e., C-score) for each pair of species in each North American Land Mammal Age (NALMA) and comparing to an effect size generated using a null model approach. The result is a series of p-values indicating whether species are aggregated (i.e., are found together more frequently than expected by chance), segregated (i.e., found together less frequently than expected by chance) or randomly associated (i.e., indistinguishable from the null model). Changes in the proportions of species association types have been used as indicators of significant ecological change, such as following the arrival of humans in North America (Lyons et al., 2016; Pineda-Munoz et al., 2020). Species co-occurrences allow for the comparison of niche similarity among and between potentially interacting taxa, and enable us to examine the factors which determine community assembly (Blois et al., 2014). Thus, we compare the niche space occupation of species pairs (aggregations, segregations and random) using estimates of their Grinellian and Eltonian niches (Grinnell, 1917; Elton, 1927; Hutchinson, 1978). The Grinellian niche is defined by the environmental conditions (e.g., habitat, temperature, precipitation) in which an organism exists, independent of competition. The Eltonian niche is constructed based on the ways in which an organism interacts with and competes with other organisms along resource use axes (Soberón, 2007; Devictor et al., 2010), and can approximated using functional traits (Chapin III et al., 2000; McGill et al., 2006; Dehling and Stouffer, 2018).

We estimate the realized Grinellian niches of species using the palynofloral record of the Bighorn Basin. Floral species richness and functional diversity covary strongly with abiotic environmental factors (Mosbrugger and Utescher, 1997; Mosbrugger et al., 2005; Jackson and Blois, 2015). Angiosperm leaf morphology is useful as a proxy for paleotemperature and mean annual precipitation (Wolfe, 1979; Wolfe, 1995) and has been used specifically in the context of the Paleocene-Eocene Bighorn Basin (Fricke and Wing, 2004). By extension, changes in the taxonomic composition of floral communities are indirectly indicative of changes in paleoenvironments (Harbert and Nixon, 2015; Harbert and Nixon, 2018; Bashforth et al., 2021). Furthermore, mammals, which maintain relatively constant body temperatures, interact most directly with plant communities and their communities are shaped by surrounding plant biomes (Bond et al., 1980; MacCracken et al., 1985; Louys et al., 2011; Suchomel et al., 2014; Luiselli et al., 2015). Furthermore, mammalian species may show strong habitat associations (e.g. Mares and Willig, 1994; Martin and McComb, 2002; Stephens and Anderson, 2014). Thus, the plant communities represent a better measure of the habitats with which mammals most closely interact and a proxy for their environments. As such, variation in mammal community richness is often best predicted by measures

of mean annual precipitation, which can also predict plant community richness (Currie, 1991, 2001; Francis et al., 2003). By quantifying the diverse compositions of palynofloral communities, and associating those compositions with mammal occurrences, we thus approximate the realized Grinellian niches of mammal species based on the range of plant communities with which they co-occur. We herein refer to those ranges of plant community co-occurrences for each mammal species as their habitat or environmental preferences (sensu Beyer et al., 2010), as they approximate the occupied subset of the sample of available environments. By comparing the realized Grinellian niches of species pairs, we can assess the degree to which environment drove changes in patterns of species occurrence.

We then estimate a component of the Eltonian niches of species using body mass. Body mass is a fundamental mammalian trait, which is a major determining factor in a wide range of other niche characteristics (Peters, 1983; Brougham and Campione, 2020). Furthermore, the degree to which species overlap in body mass trait space appears to reflect community assembly mechanisms (Bowers and Brown, 1982; Brown, 1995; Lyons and Smith, 2013; Fraser and Lyons, 2017; Pineda-Munoz et al., 2020). Mammalian body mass also covaries with other functional traits such as locomotion, thermoregulation and life history (Western, 1979; Dobson and Oli, 2007; Sibly and Brown, 2007; Lovegrove and Mowoe, 2013; Sandel, 2013; Kohli and Rowe, 2019). The diversity of traits that covary with body mass means that body mass distributions can be explained by a wide variety of competing factors, making it difficult to infer specific mechanisms affecting community assembly from body mass alone but conversely makes body mass a useful metric for broadly estimating niche space occupation and niche breadth (Grossnickle, 2020; Slater, 2022). Functional traits such as diet and locomotion can also vary widely among mammals with similar body masses (e.g., a 130-kg lion and a 130-kg wildebeest have greatly dissimilar diets and locomotor modes, despite their

comparable masses). As such, ecological results on the basis of body mass, while likely reflective of broader trends in Eltonian niche space, are not guaranteed to be same as those derived from other functional traits. Given the rarity with which functional traits can be confidently and precisely assigned from the PETM mammalian record, body mass represents the broadest and most descriptive available metric for describing their niches.

Combining Grinnellian and Eltonian methods of niche estimation provides a useful basis for the examination of ecological change through the PETM. By using multiple measures to diversely describe the mammalian niche, we can investigate how changes in functional diversity and environmental and habitat constraints, either independently or in concert, might affect community assembly. This mix of methods may hopefully provide a template for more holistic studies of community ecology in the fossil record going forward.

## Methods

*Species occurrence and ecological data.* Mammal data were acquired from 173 mammal species across 126 sites from the Paleocene and Eocene of the Bighorn Basin in Wyoming (Figure 1). The relatively limited geographic range of the localities ensures that we are examining community-scale assembly processes rather than broader continental-scale trends. The sites spanned three time bins correlated to biozones of the Clarkforkian and Wasatchian North American Land Mammal Ages (NALMAs) (Rose, 1981; Robinson et al., 2005; Secord et al., 2006), the Paleocene Clarkforkian 3 (ca. 56.2–55.8 Ma) and Eocene Wasatchian 0 (ca. 55.8–55.7 Ma) and Wasatchian 1–2 (ca. 55.7–54.8 Ma). The Wasatchian 0 encompasses the PETM. The Wasatchian 1 and 2 are combined, following Rankin et al. (2015) and Fraser and Lyons (2020). Mammal occurrences were downloaded from the Paleobiology Database using the group name "mammalia'" and the following parameters: Paleocene

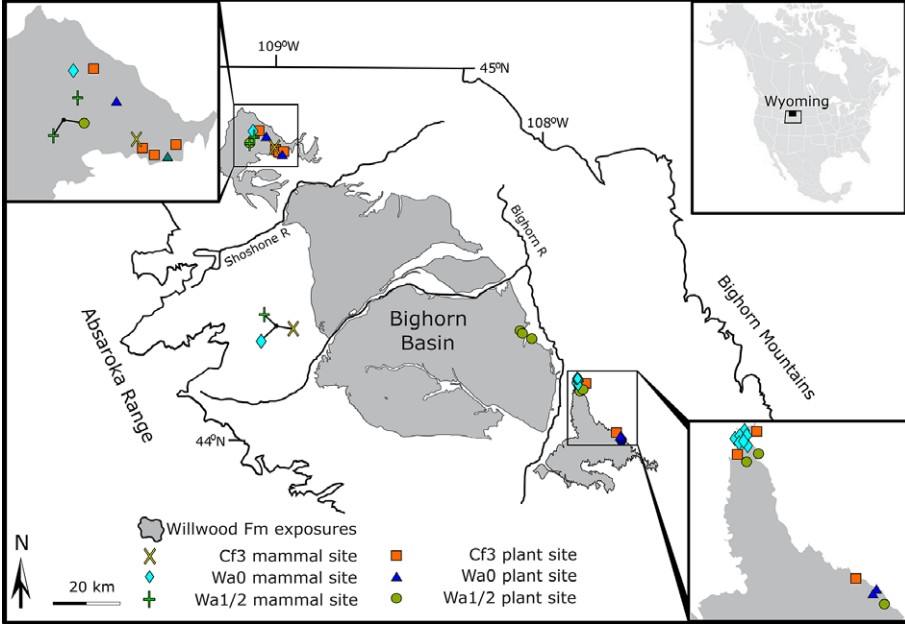

**Figure 1.** Map of the Bighorn Basin in Wyoming showing the locations of palynofloral- and mammal-bearing sites used in this study. Mammal sites are represented by three symbols corresponding to North American Land Mammal Ages: an X (Clarkforkian 3), a diamond (Wasatchian 0) and a plus sign (Wasatchian 1–2). Palynofloral sites are also represented by three symbols corresponding to North American Land Mammal Ages: a square (Clarkforkian 3), a triangle (Wasatchian 0) and a circle (Wasatchian 1–2). Black dots represent coordinates with multiple overlapping localities, which are indicated by attached black lines. Northwestern and Southeastern portions of the Basin are magnified for the sake of clarity. Exact locality information is provided in the Supplementary Information B. Base map adapted from Baczynski et al. (2013).

and Eocene; region: North America; paleoenvironment: terrestrial (data sources available from the supplementary citation list of Fraser and Lyons, 2020). Mammalian site ages are defined by NALMA subdivisions following Gingerich et al. (1980), Gingerich (1989) and Gingerich (2001). For all mammals, taxonomy was standardized to the Paleobiology Database unless literature searches indicated names had been updated. Most importantly, we made significant efforts to account for splits and synonyms, which, unlike a name change (i.e., the name of a taxon is changed but the number of total species is unchanged), could affect the results of our various analyses.

*Approximating mammal environmental niches.* Palynological occurrence data were assembled by Korasidis and Wing (2023) from sites assigned to the Clarkforkian 3 (8 sites), Wasatchian 0 (13 sites) and Wasatchian 1 (9 sites) biozones from the Bighorn Basin, Wyoming. These sites were correlated to biozones by Korasidis et al. (2023). Palynofloras were assigned to the Clarkforkian 3 (palynofloral zone P6 of Nichols and Ott [1978]) based on the presence of *Aesculipollis wyomingensis, Caryapollenites inelegans, Caryapollenites wodehousei, Echitricolpites supraechinatus, Eucommia leopoldae, Intratriporopollenites vescipites, Momipites wyomingensis, Pistillipollenites mcgregorii* and *Rousea crassimurina* in addition to dominant *Caryapollenites veripites*. Palynofloras were assigned to the Wasatchian 0 (palynofloral zone E-0 of Korasidis et al. [2023]) based on the presence of *Friedrichipollis geminus, Platycarya platycaryoides, Retistephanocolporites pergrandis, Striatopollis calidarius* and *Striatricolporites astutus* and dominant *Arecipites* spp. and *Aesculiidites circumstriatus*. Palynofloras were assigned to the Wasatchian 1 (palynofloral zone E of Nichols and Ott (1978) and Korasidis et al. [2023]) based on the presence of *Platycarya platycaryoides* and *Intratriporopollenites instructus.*

A Jaccard distance matrix was constructed from the species composition of the palynofloral sites and examined using the Nonmetric Multi-Dimensional Scaling (NMDS) and PERMANOVA functions in PAST v4.13 (Hammer et al., 2001) to establish the similarity of palynofloral assemblages among sites. NMDS and PERMANOVA analyses of palynofloral sites were also conducted on the basis of a Cosine similarity matrix, producing near-identical results (see Supplementary Information A).

Environmental preferences for mammal species were quantified using their spatiotemporal associations with palynofloral assemblages. For each of the 126 mammal sites used, the geographically closest contemporaneous palynofloral site was determined using the earth.dist function in the R package fossil (Vavrek, 2011). Each mammal site was then assigned the same NMDS scores (NMDS 1 and 2) as the closest contemporaneous palynofloral locality. Absent greater stratigraphic detail, mammal and palynofloral sites from the same NALMA subdivisions are herein considered contemporaneous, thus limiting the temporal scale of results to that of the biozone. Each species was assigned an average of the NMDS scores for the sites at which they were found, thus allowing for a species-scale comparison of environmental preference across time bins. Mean environmental preferences were calculated for each time bin and compared between time bins via the ANOVA function in PAST v4.13 (Hammer et al., 2001).

*Mammal ecomorphology.* The niche space occupation of Paleocene-Eocene Bighorn Basin mammals was inferred via body mass. Body mass estimates are as in Fraser and Lyons (2020), which were compiled from Alroy (1998), Tomiya (2013), Smits (2015) and Smith et al. (2018) and ln-transformed. Body masses were based on specimen measurements and averaged within genera where direct measurements were unavailable. Mean body masses were

calculated for each time bin and plotted through time. Dietary and locomotor inferences were not included in this analysis of mammalian niche space, due to a rarity of clear associations between dental morphology and diet in Paleocene mammals, and similarly rarity of postcranial elements appropriate for locomotor inferences.

*Species co-occurrence.* Mammal assemblages from all three biozones were subjected to a PAIRS analysis using the R package cooccur (Griffith et al., 2016) following the methodology of Ulrich (2008), Blois et al. (2014), Lyons et al. (2016) and Pineda-Munoz et al. (2020). Briefly, we determined whether pairs of species were significantly aggregated (i.e., found together more frequently than by chance), segregated (i.e., found apart more frequently than by chance) or randomly associated within each biozone. To do so, we calculated a scaled C-score for each pair of species in the site-by-species occurrence matrix for each biozone. We used the following method of calculating C-scores: $C_{ij} = (R_i − D)(R_j − D)/R_iR_j$, where $C_{ij}$ was the C-score for species pair i and j, $R_i$ was the row total (the number of species occurrences) for species i, $R_j$ was the row total for species j and D was the number of shared sites in which both species are present. For each species pair, C-score values range from 0.0 (complete aggregation) to 1.0 (complete segregation). To determine whether species were significantly aggregated or segregated, we calculated p-values, by constructing a null distribution of C-scores for each biozone by shuffling matrix elements, keeping row and column totals. Maintaining row and column totals ensures that differences in species occupancy (row totals) and sampling intensity between sites (column totals) are incorporated into the null distribution of C-scores for each species pair (Gotelli, 2000; Ulrich and Gotelli, 2010). Lyons et al. (2016) and Tóth et al. (2019) also showed that changes in the numbers of aggregated, segregated and randomly associated species pairs can be biased as a result of differences in collection mode, temporal grain, taphonomic biases, taxonomic resolution, differential sampling of abundant and rare species or differences in the spatiotemporal extents among time bins and localities, but that sampling biases can be mitigated by rigorous resampling (e.g. Tóth et al., 2019) or comparison to a null model (e.g. Lyons et al, 2016). We control for potential sampling and taphonomic biases by employing a null model, as described below. The R code used to produce the species pairs and null model is available here: https://github.com/danielleleefraser/PETMpairs.

*Ecological differences.* To test whether the ecomorphology and environmental preferences of species pairs changed through time, we followed a very similar methodology to Pineda-Munoz et al. (2020). For environment and body mass, we calculated absolute differences for all aggregated, segregated and randomly associated species pairs and took the average within each type for each biozone. Significant changes through time were assessed using randomly assembled mammal assemblages, as described below.

*Null modeling.* In the present study, we address whether the differences in environmental or habitat preferences and ecomorphology among significantly aggregated and segregated species pairs changed across the PETM. Linear regression cannot be used due to sample size limitations ($n = 3$ time bins) and a lack of statistical power. Furthermore, the metrics we employ herein may be sensitive to sampling intensity (i.e., number of samples, number of species and occupancy of sites) (Gotelli and Colwell, 2011; Ulrich et al., 2018). Thus, to assess the significance of change through time, we use a null model that randomizes the assignment of species among sites across all three biozones (Gotelli, 2000), preserving richness differences among sites and, thus, taphonomic differences. It thus randomizes patterns of species associations. We consider

this the appropriate null model for the present study due to our use of a co-occurrence metric, which calculated significant aggregation or segregation based on species occurrences across multiple sites. We then compared our observed environmental preferences and body masses to the null model using standardized effect sizes, specifically Cohen's D ($d$ = (mean observed - mean null)/standard deviation of null). We considered absolute values of $d \leq 0.2$ small effect sizes (i.e., nonsignificant differences) and $d \geq 0.8$ large effect sizes (i.e., significant differences). The above null modeling approach also addresses the role of taphonomy because richness differences were preserved among sites, thus controlling for simple taphonomic biases that could generate heterogeneity in the number of species per site and number of sites per time bin.

## Results

The total number of nonrandom and random species pairs increased from the Clarkforkian 3 to Wasatchian 0 (Figure 2), reflecting the increase in faunal species richness during the PETM (Woodburne et al., 2009; Chew and Oheim, 2013). The majority of significant pairs were aggregations; segregations were rare during the entire interval, consistent with previous results (Lyons et al., 2016; Pineda-Munoz et al., 2020). No segregated species pairs were discernible during the Wasatchian 0 biozone, precluding clear patterns from being discerned between segregated pairs.

The NMDS constructed from the 30 palynofloral localities revealed three distinct clusters of palynofloral communities along two NMDS axes. NMDS results returned a stress value of 0.1591, below the generally accepted stress limit of 0.2 in ecological research (Kruskal, 1964; Clarke, 1993; McCune and Grace, 2002). The three clusters of palynofloral communities correspond to each of the three studied time bins, these being the pre-CIE (Clarkforkian 3), CIE (Wasatchian 0) and post-CIE (Wasatchian 1) (Figure 3). These clusters showed minimal overlap in NMDS space, with all three showing significantly different compositions when analyzed by PERMANOVA (10,000 permutations, Bonferroni corrected

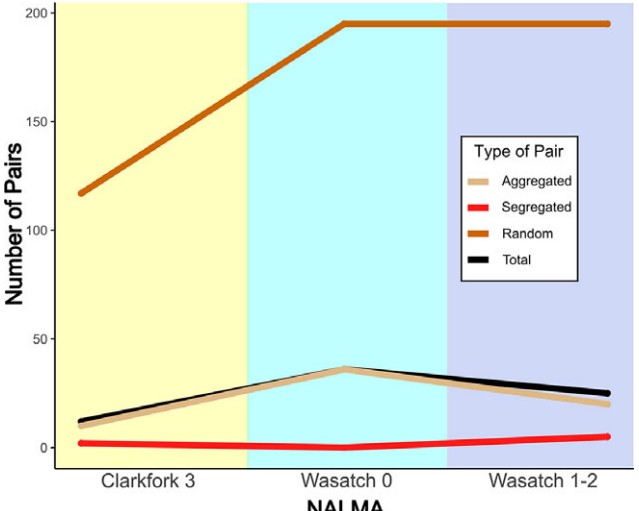

**Figure 2.** Quantities of mammal species pair types through the Clarkforkian 3 (ca. 56.2–55.8 mya), Wasatchian 0 (ca. 55.8–55.7 mya) and Wasatchian 1–2 (ca. 55.7–55.2 mya) biozones of the Bighorn Basin. Shown are numbers of significantly aggregated and segregated species pairs for each time bin, as determined through PAIRS analysis in the R package cooccur (Griffith et al., 2016). Total significant pairs (aggregated + segregated) and pairs that show no significant association or dissociation patterns (labeled here as "Random") are also shown.

p-value differences between each cluster $\leq$ 0.0003). Pairwise F-values derived from PERMANOVA are as follows: Clarkforkian 3/Wasatchian 0 = 10.89, Clarkforkian 3/Wasatchian 1–2 = 5.608 and Wasatchian 0/Wasatchian 1–2 = 13.13. The Clarkforkian 3 and Wasatchian 1–2 biozones were almost exclusively differentiated from each other along NMDS coordinate 2, while the Wasatchian 0 was differentiated from both of the other two time bins primarily along NMDS coordinate 1. These results are nearly identical to those produced on the basis of a Cosine similarity matrix (see Supplementary Information A).

The mammal localities are limited in their geographic spread (Figure 1). As such, the 126 mammal sites were most closely geographically correlated to only five contemporaneous palynofloral localities. The Wasatchian 1–2 mammal sites were geographically associated with only one palynofloral site, while the Clarkforkian 3 and Wasatchian 0 mammal sites were associated with only two different palynofloral sites each (see Supplementary Information B). However, mammal species that spanned more than a single biozone were associated with multiple palynofloral sites, which were used to inform calculations of their Grinnellian niche.

The environmental preferences, as calculated using the average of the NMDS scores of associated palynofloral sites, among aggregated species are less similar than expected under a null model during the Clarkforkian 3 for both NMDS axes, increasing in dissimilarity between the Clarkforkian 3 and Wasatchian 0 time bins to fall in-line with null expectations (Table 1; Figure 4a,b). Differences in environmental preference between segregated species are not distinguishable from null expectations for NMDS coordinate 1, but significantly decrease with respect to palynofloral NMDS coordinate 2 (Figure 4c,d). The lack of change between segregated species along NMDS coordinate 1 is expected, as that axis primarily serves to describe the uniqueness of floral communities in the Wasatchian 0 time bin (Figure 3), a time bin in which there were no significantly segregated species pairs. Thus, differences in environmental preference among segregated species pairs are best described with respect to NMDS coordinate 2. Differences in environmental preference between randomly paired mammal species all fall within the standard deviation of the null model for both NMDS coordinates (Figure 4e,f). Across all species, mean environmental preferences also appear to change significantly between time bins along both NMDS coordinates (Figure 4g), with ANOVA results showing highly significant (p < 0.0005) Mann–Whitney pairwise differences in both NMDS values among all time bins with the exception the Wasatchian 0 and 1/2 time bins, which could be differentiated along NMDS coordinate 1, but not along NMDS coordinate 2 ($p$ = 0.6636); these trends appear the same when means for each time bin are calculated only from taxa whose first occurrences fall within that time bin (Figure 4i–j). The range of mammalian environmental preferences appears expand from the Clarkforkian 3 (NMDS1 = 0.02219, NMDS2 = 0.0776) to the Wasatchian 0 (NMDS1 = 0.02576, NMDS2 = 0.100) and Wasatchian 1–2 (NMDS1 = 0.02587, NMDS2 = 0.0831), as appears to be the case in both the total set of mammalian taxa and the set only including first occurrences (Figure 4g–j). These results, combined with the palynofloral assemblage data, show an increase in the difference in environmental preference among commonly associated mammalian species over the studied interval in conjunction with progressive change in the environmental setting. That change is also accompanied by a greater similarity in the environmental preference among segregated species.

Mammalian species pairs showed minimal changes in the body mass components of Eltonian niche occupation (Figure 5). We

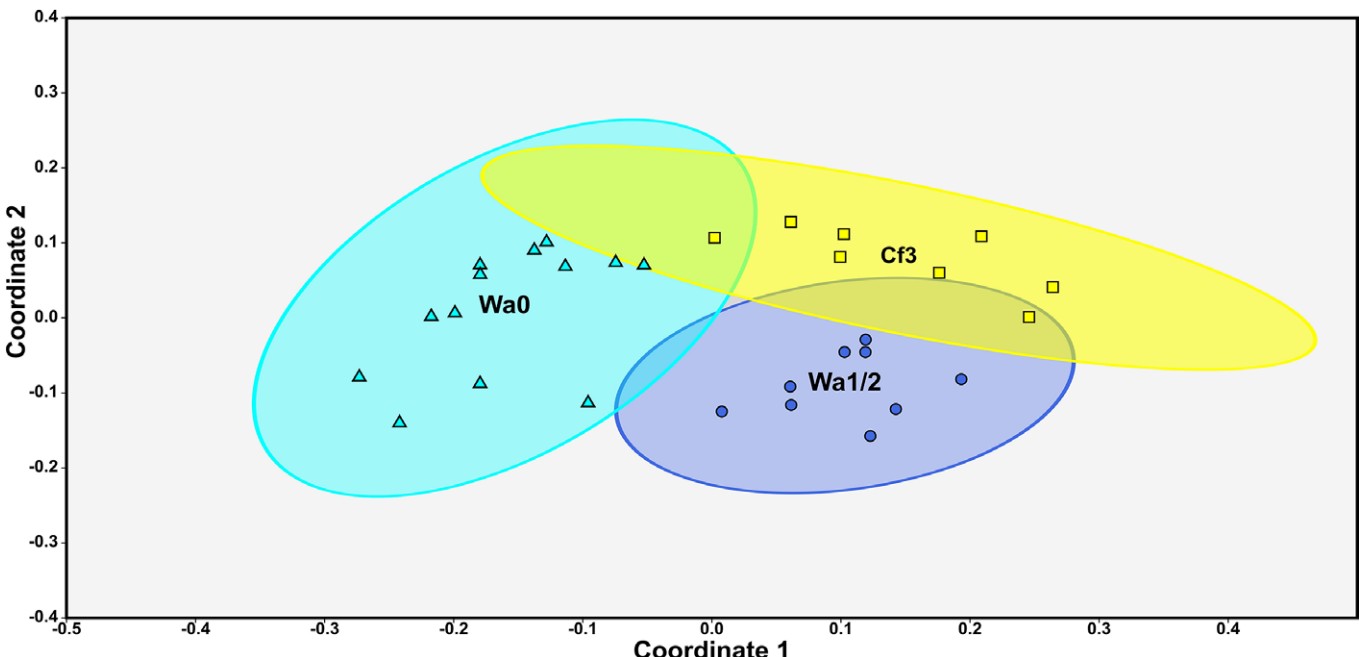

**Figure 3.** NMDS plot displaying the similarity in composition of palynofloral communities from each of the Clarkforkian 3 (Squares, Cf3), Wasatchian 0 (Triangles, Wa0) and Wasatchian 1–2 (Circles, Wa1–2) time bins. NMDS was produced based on a Jaccard similarity matrix comparing the taxonomic occurrences of all 30 palynofloral sites. NMDS is ordinated across 2 axes, returning a stress value of 0.1591. Palynofloral assemblage age assignments derived from Korasidis et al. (2023) and presence-absence data from Korasidis and Wing (2024). Ellipses represent 95% confidence intervals of the total assemblage space occupied in each time bin.

**Table 1.** Effect sizes of differences in climate preference between species pairs within each of the Clarkforkian 3 (ca. 56.2–55.8 mya), Wasatchian 0 (ca. 55.8–55.7 mya) and Wasatchian 1–2 (ca. 55.7–55.2 mya) biozones. Differences in climate preference are shown along each of the 2 palynofloral NMDS axes. Differences are shown separately among pairs of each type: aggregated (Agg.), segregated (Seg.), and random (Rand.). Segregated species pairs are not observed from the Wasatchian 0 biozone

| Biozone | Agg. Diff. (NMDS1) | Seg. Diff. (NMDS1) | Rand. Diff. (NMDS1) | Agg. Diff. (NMDS2) | Seg. Diff. (NMDS2) | Rand. Diff. (NMDS2) |
|---|---|---|---|---|---|---|
| Clarkforkian 3 | −1.042142 | −0.951861 | −0.936426 | −1.035858 | −0.682449 | −0.937878 |
| Wasatchian 0 | −0.395042 | NA | −0.195936 | −0.760942 | NA | −0.721962 |
| Wasatchian 1–2 | −0.551791 | −0.845517 | −0.606639 | −0.653171 | −0.953286 | −0.501387 |

observe no change in body mass differences regardless of the type of species pair through the PETM (Figure 5a–c). We observed a slight, but statistically insignificant decrease in mean body mass across all taxa between the Clarkforkian 3 and Wasatchian 0 biozones, with ANOVA results showing no significant Mann–Whitney pairwise differences (p-values >0.61) in mean log body mass among any of the three time bins (Figure 5d), a pattern that is repeated (p-values >0.48) when looking only at the first occurrences (Figure 5e).

## Discussion

The PETM in North America was characterized by rapid, short lived climate change, the arrival of Eurasian (i.e., artiodactyls, perissodactyls and primates) immigrants, and northward range expansions of endemic mammals that were not balanced by extinctions (Bowen et al., 2002; Gingerich, 2006; Burger, 2012; Fraser and Lyons, 2020). The results were richer mammal communities (i.e., greater γ and α diversity) and, potentially, changes in how communities were assembled (Burger, 2012; Fraser and Lyons, 2020). Today, the assembly of tropical and temperate mammal communities are driven by divergent processes; temperate mammal communities are subject to environmental filtering, a process

whereby species are sorted along abiotic and biotic gradients according to their environmental tolerances (Weiher et al., 1998; Soininen et al., 2007a; Soininen et al., 2007b; Kraft et al., 2015), to a greater degree than tropical communities (Hawkins et al., 2003; Currie et al., 2004; Helmus et al., 2007). The assembly of tropical mammal communities may be driven more by species–species interactions (i.e., resource competition) than their temperate counterparts, though this pattern may not always hold true (Safi et al., 2011; Fraser and Lyons, 2017). We therefore expected significant changes in the assembly of PETM mammal communities. The methods used herein can be useful tools for teasing apart the processes driving mammalian community assembly through the PETM (Blois et al., 2014; Tóth et al., 2019). Through identifying aggregated and segregated species pairs, we make fruitful comparisons of body masses and environmental preferences that enhance our ability to differentiate assembly processes such as environmental filtering and resource competition (Blois et al., 2014; Tóth et al., 2019; Pineda-Munoz et al., 2020).

The PETM was a rapid climate change event that may have changed the degree to which environmental filtering drove community assembly. Under an environmental filtering model, we expect species with similar environmental preferences to inhabit

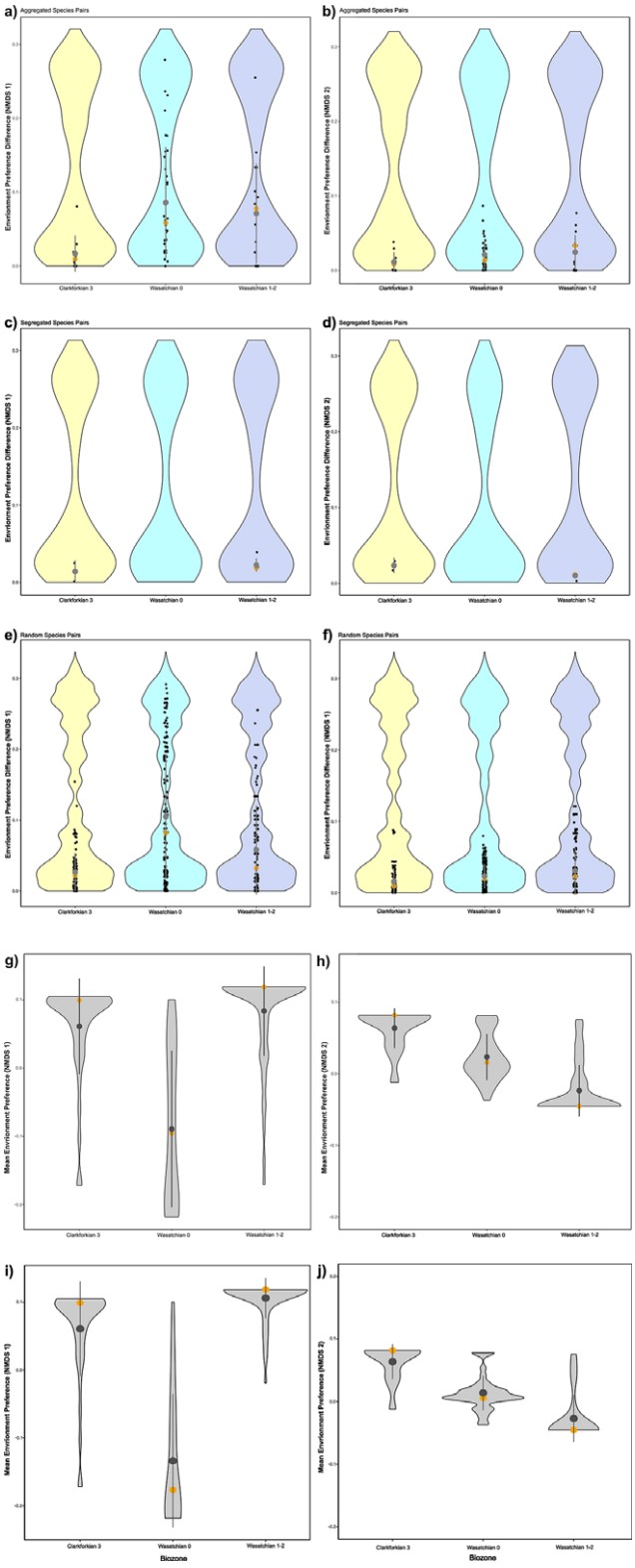

**Figure 4.** Mean differences in environmental preference between mammal species pairs and regional environmental preference distributions through the Clarkforkian 3 (ca. 56.2–55.8 mya), Wasatchian 0 (ca. 55.8–55.7 mya) and Wasatchian 1–2 (ca. 55.7–55.2 mya) biozones. Environmental preferences are here determined as a mean of the NMDS coordinates of the closest contemporaneous palynofloral sites for each mammal taxon. Mean and median differences in scores along NMDS coordinate 1 are shown for aggregated (a), segregated (c) and random (e) species pairs. Mean and median differences in scores along NMDS 2 coordinate 2 are likewise shown for aggregated (b), segregated (d) and random (f), species pairs No differences in environmental

preference are shown between segregated species pairs in the Wasatchian 0 biozone as there were no significantly segregated species pairs determined from that interval. Differences in environmental preference values which fall within the expectations of our null model are shown by the colored violin plots. Distributions of environmental preferences among all Bighorn Basin mammals are shown in gray with means and medians along NMDS 1 (g and i) and NMDS 2 (h and j). We here show both mean palynofloral NMDS scores of all mammal taxa occurring within a given time bin (g and h), and mean scores for each time bin of only those mammal taxa which first occur in that time bin (I and J). In all cases, mean differences are shown as gray dots, while median differences are shown as orange dots, with 95% confidence intervals shown as gray bars. Regional environmental preference distributions were not compared to a null model.

the same communities (Weiher et al., 1998; Kraft et al., 2015). Furthermore, those species should share traits that mediate their relationship with the environment (Weiher et al., 1998; Cornwell et al., 2006), in this case, body mass. Body mass may directly interact with climate through the laws of thermodynamics (Porter and Gates, 1969; Ahlborn, 2000; Gillooly et al., 2001) but is also correlated with many other fundamental components of mammalian niches (Western, 1979; Peters, 1983; Iriarte-Diaz, 2002; Dobson and Oli, 2007; Sibly and Brown, 2007; Lovegrove and Mowoe, 2013; Kohli and Rowe, 2019). In the context of species pairs, under an environmental filtering scenario, significantly aggregated species should be most similar in associated floral NMDS scores and body mass, while segregated species should be least similar in both categories (Blois et al., 2014).

We show that environmental preferences among aggregated species pairs were more similar than null expectations during the Clarkforkian 3 and that differences in environmental preferences became indistinguishable from the null during and after the PETM (Table 1; Figure 4a,b), which might indicate that community assembly may no longer have been driven by environmental filtering (sensu Blois et al., 2014). However, we find the same pattern for random and segregated (along NMDS 1) species pairs (Table 1; Figure 4a, b), suggesting that the observed change among aggregated pairs likely does not indicate a change in community assembly. Conversely, we show that differences in environmental preferences among segregated species along NMDS 2 were indistinguishable from null expectations until the Wasatchian 1–2 biozones, whereupon they become significantly similar (Figure 4d), evidencing a lack of an environmental filter with respect to paleofloral habitats. We also find a lack of significant change in comparative body masses among species pairs of all types (Figure 5a–c), in agreement with Fraser and Lyons' (2020) observations of static body mass dispersion through the same period. While we can only reject post-Clarkforkian environmental filtering of mammalian taxa through the lens of palynofloral communities, there is a strong correlation between changes in floral community composition and changes in abiotic environmental variables (e.g., mean annual precipitation, temperature and seasonality), both in general (Laughlin et al., 2011; Harbert and Nixon, 2015; Harbert and Nixon, 2018; Bashforth et al., 2021; Bricca et al., 2022) and in the specific case of the PETM in the Bighorn Basin (Fricke and Wing, 2004; Wing and Currano, 2013; Korasidis and Wing, 2023). Given the breadth of associations between plant community composition and the abiotic environment, we consider associations between mammals and palynofloral assemblages to be reflective of broader mammalian environmental preferences. Furthermore, while there are estimates of MAT for the Bighorn Basin during the PETM (e.g Snell et al., 2013), mammals with their relatively constant body temperatures are expected to interact most directly with plant communities, apparent as significant correlations between

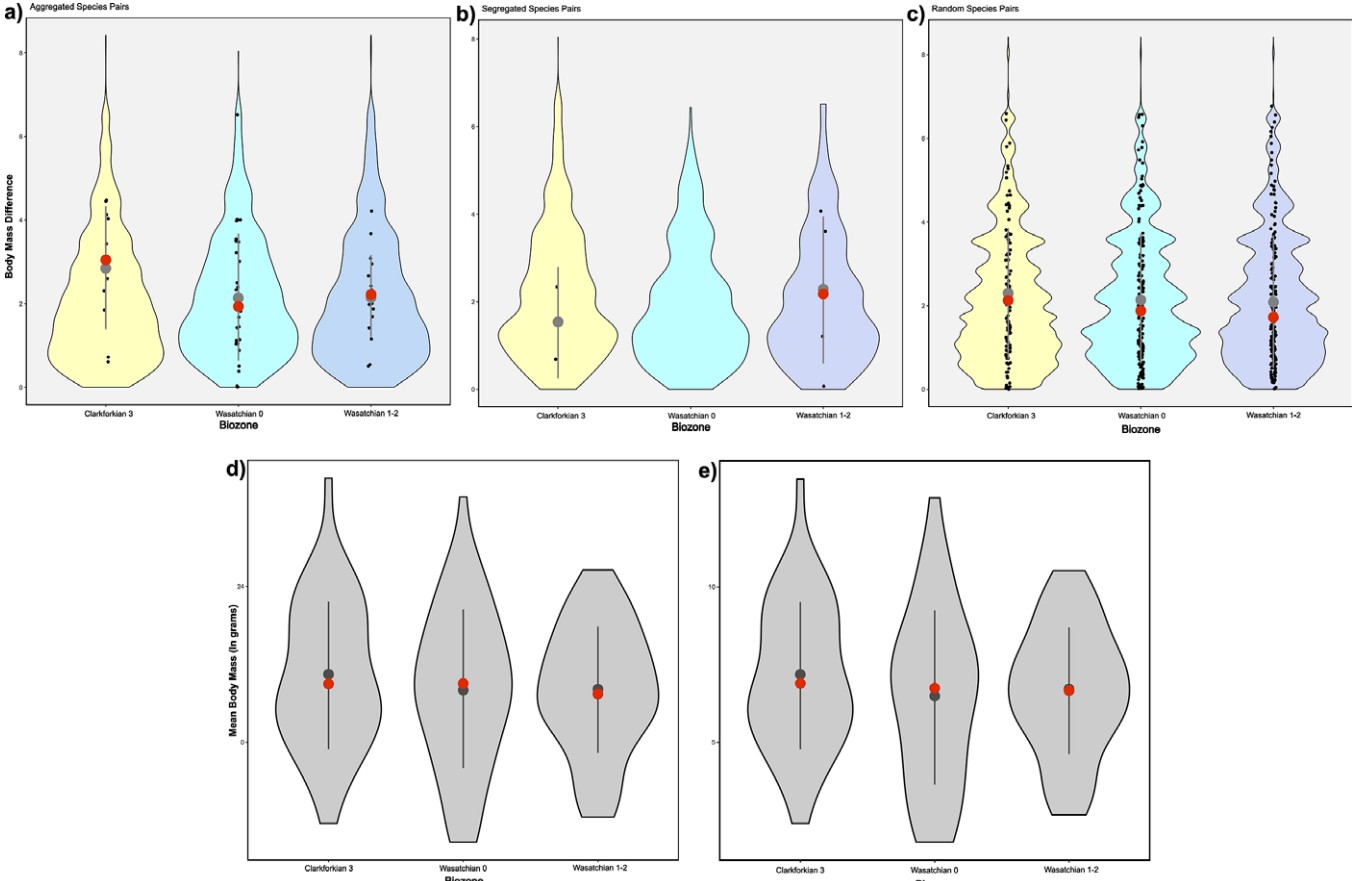

**Figure 5.** Mean differences in body mass between mammal species pairs and regional body mass distributions through the Clarkforkian 3 (ca. 56.2–55.8 mya), Wasatchian 0 (ca. 55.8–55.7 mya) and Wasatchian 1–2 (ca. 55.7–55.2 mya) biozones. Body mass estimates are here compiled from Alroy (1998), Tomiya (2013), Smits (2015) and Smith et al. (2018) and ln-transformed. Mean differences are shown for aggregated (A), segregated (B) and random (C) species pairs. No differences in body masses are shown between segregated species pairs in the Wasatchian 0 biozone as there were no significantly segregated species pairs determined from that interval. Differences in body mass which fall within the expectations of our null model are represented by colored violin plots. Distributions are shown for all body mass estimates from mammal taxa occurring within a given time bin (D), and for each time bin of only those mammal taxa which first occur in that time bin (E). In all cases, mean differences are shown as gray dots, while median differences are shown as red dots, with 95% confidence intervals shown as gray bars. Regional body mass distributions were not compared to a null model.

mammal richness and primary productivity or energy in both space and time (Currie, 1991; Fritz et al., 2016). There are also potentially unmeasured environmental variables for which correlations with PETM floral community structure are unknown, which may represent unexplored environmental filters. However, based on the analyses herein, we suggest that a change in the strength of environmental filtering was unlikely to have been a significant factor in the assembly of post-Clarkforkian mammalian communities.

A reduced role for environmental filtering in community assembly, specifically relating to the biotic and abiotic conditions created by plant assemblages, during and after the PETM could suggest either a change in the variance of mammal environmental/habitat preferences or spatial variability of environment (Peres-Neto et al., 2012; Blois et al., 2014; Daniel et al., 2019). An increase in the overall variance of preferred environments or habitats among mammal taxa (Figure 4g–h) could increase the similarity of environmental preferences among taxa, regardless of whether they are aggregated or segregated pairs, reducing the differentiability of the observed differences from the null model. Such a trend may have been a result of greater environmental generalism among newcomer taxa; taxa with first appearances in the Wasatchian 0 appear to be associated with a wider array of palynofloral assemblages (Figure 4j–i), though the same taxa only show a small apparent

increase in the variance of body masses (Figure 5e). Broader environmental preferences would have made the newcomer taxa more likely to successfully shift their ranges in a period of climatic upheaval like the PETM (Bergman, 1988; Parsons, 1994; Marvier et al., 2004; Richmond et al., 2005). However, taxa that survived through the Clarkforkian 3 into Wasatchian 0 also tended to possess wide environmental preferences (Figure 4g–h), likely reflecting persistence of taxa that could weather PETM climate change, as has been modeled and observed in environmental generalists with respect to modern climate change (e.g. Warren et al., 2001; Juilliard et al., 2004; Thomas et al., 2004).

Stability of individual species' Grinnellian niches (i.e., association with palynofloral assemblages) through time is an assumption that has allowed for the prediction of historical ranges (Svenning et al., 2011) and responses to climate change (e.g. Peterson, 2003; Tingley et al., 2009). Individual taxa can, however, undergo significant changes in their geographic distributions and, thus, environmental conditions in which they live. Over longer timescales, native taxa tend to constrict their realized Grinnellian niche space when faced with climate change and introduction of immigrant taxa (Peterson, 2011; Brame and Stigall, 2014; Stigall, 2014). Immigrant taxa also alter their Grinnellian niche occupation during range shift events over short timescales (e.g., Broennimann et al., 2007;

Early and Sax, 2014). We calculated species environmental preferences in a time transgressive manner, however. That is, environmental preferences for each taxon were calculated based on their occurrences across all three biozones, as a closer approximation of their long-term environmental preferences and, potentially, environmental tolerances. Furthermore, the spatially clustered nature of the palynofloral and mammal occurrences within each biozone means that the only way to capture variability in mammal environmental preferences is to compute means across time bins. Thus, the observed changes in mean environmental preference, variance and reduced similarity among species pairs must result from changes in species composition rather than shifts in the Grinnellian niches of individual taxa through time.

Enhanced spatial homogeneity of environment does not appear to explain the changes in the comparative environmental preferences among species pairs. Palynofloral and macrofloral communities do not appear to show significant changes in spatial variability through the PETM (Wing and Currano, 2013, Korasidis and Wing, 2023). Mammal communities also show either an increase (Darroch et al., 2014) or no change (Fraser and Lyons, 2020) in β diversity during the PETM, suggesting that enhanced environmental homogeneity did not drive the observed increase in environmental preference similarity among segregated species during the Wasatchian 1–2 biozones (Figure 4d).

Increases in mammal α and γ diversity during the PETM of North America (Gingerich, 2003; Darroch et al., 2014; Chew, 2015; Fraser and Lyons, 2020) may also have changed the competitive landscape; new arriving species, such as artiodactyls and perissodactyls, overlapped in stable isotope (i.e., $\delta^{13}C$ and $\delta^{18}O$) and, thus, dietary and environmental niche space with endemic taxa (Secord et al., 2008), potentially enhancing the likelihood of interspecific resource competition. While species that immigrated into the Bighorn Basin during the PETM tended to be smaller (Figure 5e) than incumbent taxa (Bowen et al., 2002; Gingerich, 2006), this does not on its own provide significant evidence against enhanced resource competition among species.

The outcome of competitive interactions can include competitive exclusion or limiting similarity, phenomena for which we find no evidence with respect to body masses from the Clarkforkian 3 through Wasatchian 1–2, as they were not differentiable from null expectations (Figure 5a–c). These results complement those of Fraser and Lyons (2020), who suggested that the niche space of PETM mammalian communities in North America was unsaturated based on a lack of evidence for enhanced intraspecific interactions despite increased species richness. The principle of competitive exclusion posits that species with extremely similar or identical Eltonian niches may not coexist in the same community (Hardin, 1960). Limiting similarity predicts that there be an upper limit on the degree to which coexisting species can overlap in resource use (MacArthur and Levins, 1967; Abrams, 1983). Under both limiting similarity and competitive exclusion, we expect greater differences in body mass distributions among aggregated species pairs than the null expectation, and fewer differences among segregated species pairs than the null expectation (sensu Blois et al., 2014). However, the longer temporal scales averaged by paleontological data mean that significant aggregations or segregations may be products of environmental or taphonomic factors in addition to biotic interactions (Blois et al., 2014). If neither taphonomy nor environment can explain significant aggregations or segregations of species, then we can infer that biotic interactions played a role in community assembly. In the present case, interspecific interactions can be confidently invoked if species

associations are not already explained by differences in environmental preference and if differences in body masses among significantly aggregated or segregated species differ significantly from null expectations, which address taphonomic effects (see *Methods*). Likewise, competitive exclusion, predicted to manifest as significant similarity among segregated species pairs, may be less discernible with increasing spatial scale (Araújo and Rozenfeld, 2014), though some exceptions from both mammalian and avian ecology do show larger-scale discernibility of interactions (Gotelli et al., 2010; Safi et al., 2011; Fraser and Lyons, 2017). The spatial scale of the present study (~200 km) is comparatively small, thus enhancing the potential for detecting interspecific interactions, yet we were still unable to detect any evidence of body mass–mediated changes in community assembly among PETM mammals.

While body mass can account for a large part of an animal's Eltonian niche, it is not perfectly correlated with every other Eltonian niche metric. Body mass is broadly collinear with traits such as diet and locomotion (Western, 1979; Sibly and Brown, 2007; Lovegrove and Mowoe, 2013), but these traits may exhibit different patterns to that of body mass in mammalian communities through the PETM. There are changes documented in the diversity of locomotor modes between Clarkforkian and Wasatchian mammals resulting from the arrival of diverse digitigrade and unguligrade immigrants, among others (Rose, 1990; Gould, 2017). Dietary niche occupation, too, may have seen changes during the PETM (Stroik and Schwartz, 2018; Selig et al., 2021). However, there is not yet sufficiently detailed or abundant information on the locomotor or dietary habits of PETM Bighorn Basin mammals to support co-occurrence analyses like PAIRS.

In lieu of niche space expansion, stasis in the similarity of body mass space between species pairs during a period of increasing taxonomic diversity could be explained by enhanced primary productivity as competition for resources is relaxed (MacArthur, 1972; Strobeck, 1973; Lawlor, 1980). The Bighorn Basin experienced an increase in the abundance of nitrogen-fixing legumes during the PETM (Bruneau et al., 2008; Currano et al., 2016), conditions which are associated with increased primary productivity (Epihov et al., 2017). Mean annual precipitation, also considered a correlate of primary productivity (Chapin III et al., 2011), decreased at the beginning of the PETM, but rebounded by the end of the CIE body (i.e., roughly contemporaneous with the end of the Wasatchian 0 biozone) (Wing et al., 2005; Kraus and Riggins, 2007; Secord et al., 2012). It is therefore possible that, at least by the end of the Wasatchian 0, primary productivity had increased sufficiently to dampen competition for resources, though more direct estimates of regional primary productivity are needed to test this hypothesis. Were it to have occurred, increased primary productivity would represent a reasonable mechanism for the unsaturation of mammalian niche spaces interpreted by Fraser and Lyons (2020).

Climate change and range shift events like those seen in the modern are often associated with climatic generalism, particularly on the part of incoming immigrant taxa (Bergman, 1988; Vermeij, 1991; Marvier et al., 2004). Our results from the PETM of the Bighorn Basin indicate that such events may also broaden the range of environmental preferences exhibited by subsequent post-event communities as a whole, removing environmental preference as a determining factor in community assembly and homogenizing environmental preference across geography. These results are likely best compared to modern communities that are unsaturated in niche occupation or which occur in environments with increasing primary activity. Our observations of Grinnellian niche variance also indicate that assumptions of niche stability over time may not

be applicable at the community scale, as the predictive power of environmental preference on community assembly is lost with the onset of climate change and faunal turnover.

## Conclusions

Our findings combine to depict a changing environment in the PETM of the Bighorn Basin, which features little to no change in the occupation of body mass space, despite dramatic taxonomic and environmental change. We find no evidence for changes in the degree to which community assembly was driven by environmental preferences or functional roles between the Wasatchian 0 and Wasatchian 1–2 biozones. The methods used herein demonstrate the utility of incorporating different distinct modes of niche quantification in elucidating the effects of environmental disturbance and range shifts on community structure. The decrease of differences in environmental preference between communities (shown here through segregated species pairs) additionally indicates that environmental preference was not only not a determining factor in the geographic separation of species, but that across geography species were significantly more homogeneous in their environmental preferences than expected. Given that body mass also does not appear to have an impact on community assembly during this time, we are left with no distinct drivers of species segregation. Our results appear to concur with the idea that the unsaturated nature of mammalian communities may have enabled functional stability during a period of climate change and faunal turnover.

**Open peer review.** To view the open peer review materials for this article, please visit http://doi.org/10.1017/ext.2024.25.

**Supplementary material.** The supplementary material for this article can be found at http://doi.org/10.1017/ext.2024.25.

**Data availability statement.** The data that support the findings of this study are available from the corresponding author, M.A.J.B.W., upon reasonable request. R code used to produce mammal species pairs and null models is available here: https://github.com/danielleleefraser/PETMpairs.

**Acknowledgments.** We would like to thank Dr. Scott Wing of the Smithsonian Institution National Museum of Natural History for his aid and insights. We would also like to thank Dr. Hillary Maddin and the members of her lab at the Earth Sciences Department at Carleton University for their aid and consultation.

**Author contribution.** Conception and design of work: M.A.J.B.W., V.A.K. and D.F.; Data collection: M.A.J.B.W., V.A.K. and D.F.; Data analysis: M.A.J.B.W. and D.F. Drafting and revising: M.A.J.B.W., V.A.K. and D.F.

**Financial support.** This study was conducted with financial support from the Natural Sciences and Engineering Research Council of Canada (NSERC RGPIN-RGPIN-2018-05305). Funding for the palynofloral research was provided through a Smithsonian Institution Peter Buck Postdoctoral Fellowship Award to V.A.K. V.A.K. is currently funded by the Elizabeth and Vernon Puzey Fellowship Award through The University of Melbourne.

**Competing interest.** The authors declare no competing interest.

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
