## [Editor Report]

We have now received two reviews for your manuscript; we appreciate the time delay in getting these, however, unfortunately it proved difficult securing timely peer-reviews quickly.

Both reviewers were broadly positive about your manuscript however both also had major comments to make on both your data, analyses, and interpretations. Both highlighted the issues with the ecomorphological data you use, in particular your reliance on Lovegrove & Mowoe’s posture categories. Reviewer 2 suggests some ways you could get around some of the issues with this approach, while reviewer 1 suggests removing these data altogether from your study. Reviewer 2 also suggests some consideration regarding the use of body mass as an ecomorphological category.

Reviewer 1 also made some critical and/or major analytical points regarding your statistical choices, what they mean biologically, and areas where you will need to either provide further justification or verification of your results, or consider different approaches. These are presented as ways of improving both your interpretations and discussions, and should be seriously considered in any revision.

---

## [Editor Report]

Both reviewers appreciate the changes you have made to your manuscript in light of their initial sets of comments. Their recommendations are overall positive, however, divergent in how much additional revision your manuscript may need before publication. Reviewer 1 has indicated minor revision and indeed their suggestions, while important, will not require substantial changes to your manuscript. Reviewer 2 has both continuing and new concerns regarding the statical treatment and conceptualisation of your analyses. It’s possible that these can be addressed in a straightforward way; however, on balance I’m recommending major revisions are probably necessary, and I leave open the possibility that any revised draft will be sent for further review.

---

## [Editor Report]

Reviewer 1 is satisfied that all their comments have been adequately addressed and that your manuscript can be published. Reviewer 2 was not available to evaluate your revisions, so a new reviewer was approached. Although they have recommended rejection, I find on balance that their criticisms, while valid and relevant, can be addressed with some relatively minor revisions. Their comments, and my suggestions on revision, are below:

1. Toning down the relationship between your study and modern climate change - this can be easily done in your significance statement and your introduction.

2. I completely agree that your use of the terms ecomorphospace and functional diversity are not appropriate when discussing changes in body size, even if the latter has some influence on other ecological properties of mammals. These terms can be substituted throughout the text.

3. The correlation of the palnyfloral communities and the time bins. This is an important point that should be discussed. Is species turnover and change in richness, which obviously happens in the plant communities, not simply passed on to the mammal communities, such that the latter is entirely produced by the former?

4. Estimation of MAT and MAP for each time bin might help account for the above. These should be added, or an indication of why they aren’t added provided.

5. Need to add a discussion on why your results are not simply reflecting taphonomic differences between time bins.